# Engineering Solutions for Mitigation of Chimeric Antigen Receptor T-Cell Dysfunction

**DOI:** 10.3390/cancers12082326

**Published:** 2020-08-18

**Authors:** Artemis Gavriil, Marta Barisa, Emma Halliwell, John Anderson

**Affiliations:** UCL Great Ormond Street, Institute of Child Health, London WC1N 1EH, UK; a.gavriil@ucl.ac.uk (A.G.); m.barisa@ucl.ac.uk (M.B.); emma.halliwell.17@ucl.ac.uk (E.H.)

**Keywords:** chimeric antigen receptor, cancer immunotherapy, T cell exhaustion, T cell dysfunction

## Abstract

The clinical successes of chimeric antigen receptor (CAR)-T-cell therapy targeting cell surface antigens in B cell leukaemias and lymphomas has demonstrated the proof of concept that appropriately engineered T-cells have the capacity to destroy advanced cancer with long term remissions ensuing. Nevertheless, it has been significantly more problematic to effect long term clinical benefit in a solid tumour context. A major contributing factor to the clinical failure of CAR-T-cells in solid tumours has been named, almost interchangeably, as T-cell “dysfunction” or “exhaustion”. While unhelpful ambiguity surrounds the term “dysfunction”, “exhaustion” is canonically regarded as a pejorative term for T-cells. Recent understanding of T-cell developmental biology now identifies exhausted cells as vital for effective immune responses in the context of ongoing antigenic challenge. The purpose of this review is to explore the critical stages in the CAR-T-cell life-cycle and their various contributions to T-cell exhaustion. Through an appreciation of the predominant mechanisms of CAR-T-cell exhaustion and resultant dysfunction, we describe a range of engineering approaches to improve CAR-T-cell function.

## 1. Introduction

Adoptive immunotherapy for cancer with gene-modified T-cells has become a forefront field of scientific discovery and clinical trial focus in the last 10 years. Following on from encouraging seminal studies using T-cell receptor gene-modified autologous T-cells [1], the arena is now dominated by chimeric antigen receptors (CARs). Originally conceived in the late 1980s [2,3,4], prototypic initial generation CARs for clinical evaluation comprise of an ectodomain, most commonly derived from a monoclonal antibody with specificity for a target antigen, and an endodomain, comprising T-cell signalling motifs that become activated following antigen binding and the formation of a synthetic immunological synapse (Figure 1) [5]. Trials of CAR-T cells targeting the B cell antigen CD19 have shown remarkable sustained complete responses in patients with refractory relapsed B cell cancers, although with associated toxicity [6,7]. Solid tumours meanwhile have been generally refractory for diverse reasons likely involving a composite of physical barriers to access of T-cells, the presence of suppressive immune cells and tumour cells, the variability of expression of the target antigen, and (in many tumours) low mutational burden limiting antigen spreading and generation of memory responses [8,9,10].

A growing consensus further points to T-cell exhaustion and/or dysfunction as a cause for CAR-T therapeutic failure. The idea that this exhaustion may be prevented or even reversed is gaining traction, and may offer a potential solution to the lack of adequate clinical responses in the solid tumour context. Understandable enthusiasm for targeting the exhaustion phenotype in CAR-T-cells has been generated by the striking clinical success of immune checkpoint inhibitors targeting, firstly, the CTLA-4 pathway, and subsequently, the PDL1/PD1 pathway. However, firm translational an understanding of the relationship between CAR-T-cell functionality and the emergence of exhaustion/dysfunction requires an understanding of the physiological role of T-cell exhaustion in the framework of the transcriptional and epigenetic regulation of T-cell development. 

The term “T-cell exhaustion” was first introduced to describe T-cell hypo-functionality in the context of chronic viral infection in mice [12,13,14]. Subsequently, the phenomenon has been broadly applied to a range of infectious agents and cancers [15,16,17]. In the context of chronic infection, a rational understanding of “exhausted” T-cells is of a population which persists but has dampened effector function, and thus, has an important function in limiting infection whilst also limiting the immunopathology of an unrestrained T-cell immune response. Recent work on the development of the exhausted phenotype suggests a parallel T-cell developmental pathway which distinguishes exhausted cells from classical effectors [14,18,19,20]. The latter undergo contraction by anergy following acute antigen exposure, leaving a small population of memory cells to effect long term protection [21]. In contrast, a population of “exhaustion precursor cells” (also referred to in recent literature as the progenitor, stem/memory-like or follicular cytotoxic cells) is resistant to anergy and come to dominate during chronic antigen stimulation by virtue of surviving the contraction stage (Figure 2) [22,23,24]. The hypothesis that tumour infiltrating T-cells expressing classical exhaustion markers, such as PD1 and TIM3, represent the same developmental pathway as exhausted cells in chronic injection, is an area of intense research interest. Such physiological context may be highly relevant to the CAR-T solid tumour field where a homogenous population of T-cells stimulated by a highly expressed target antigen are likely to be subjected to ongoing antigenic stimulation in a manner that is analogous to the setting of chronic infection. 

### 1.1. The Molecular Regulators of Exhaustion

The elucidation of several key genes that control and regulate states of T-cell exhaustion from transcriptional and epigenetic studies has confirmed the notion that exhaustion and effector developmental pathways are parallel processes, which have evolved to allow a more dynamic range of possible T-cell responses. The effector pathway involves highly active, but short-lived bursts of adaptive killing, whereas the exhaustion pathway induces a population of T-cells with diminished effector function while maintaining the capacity for more sustained persistence (Figure 2).

A recent Viewpoint article from 18 leaders in the field [18] indicates that, whilst there are areas of ongoing controversy, a consensus is emerging based on mechanistic evidence of the molecular and developmental pathways by which effector and exhaustion phenotypes are related. The principal tenet is of a population of exhaustion precursor cells that, in common with effector precursors, require TCF-1 for their establishment and have a gene expression signature—including, but not restricted to, *Il7r*, *Bcl6*, *Id3*, and *Lef1* [20,25]. Whereas, effector progenitors differentiate to effector cells characterised by expression of KLRG1, the PD-1^int^ TCF1^+^ exhaustion progenitors become PD1^hi^ TIM-3+ TCF1-cells, variously referred to as terminally exhausted or dysfunctional [18], due to their limited capacity for effector function, and high expression of CD38, CD101, LAG3, and TIGIT. The term precursor has been championed in preference to progenitor or stem for both exhaustion and effector pathways, since the cells are already established on a differentiation pathway and may have limited differentiation potential [19]. Early work using adoptive transfer of cells with the exhaustion phenotype has demonstrated that this subpopulation can survive long-term and mount a recall response to antigen [26]. Several investigators have subsequently reported that it is the exhaustion precursor pool that is self-renewing and responsible for clinical responses to PD1 pathway blockade [27,28]. 

Studies in tumour-bearing mice have demonstrated that the exhaustion pathway is established early during tumourigenesis [29], has an epigenetic signature distinct from the effector pathway [30], and is established and maintained by the action of the transcription factor and epigenetic modifier TOX [31,32,33,34,35,36,37,38]. Several lines of evidence point to TOX being activated following TCR engagement to NFAT mediated transcription (Figure 3). Once expressed, TOX hardwires T-cells into the exhaustion phenotype through epigenetic modification (e.g., interaction with the H3 and H4 acetylation complex HBO1 [31]) and regulation of other proteins driving exhaustion, such as the transcription factor NR4A [39,40] and the type 1 transmembrane protein SLAMF6 [41].

The balance between exhaustion and effector function, although epigenetically hardwired, is phenotypically more dynamic with the interaction of NFAT and AP1 transcription factors being critical. NFAT is activated by dephosphorylation following TCR or CAR engagement and binds the promoter of target genes. By forming complexes with classical AP1 heterodimers of JUN and FOS, effector genes, such as IL-2, are transcribed. In the absence of AP1, or in overactivation of NFAT in a highly activated T-cell, NFAT directs a transcriptional signature of genes that induce exhaustion. Three patterns of NFAT occupancy of target promoters can be envisioned: (1) Classical NFAT/AP1 dimers drive transcription of effector genes, whilst (2) NFAT dimerised with alternate bZIP members (e.g., JUNB or IRF4) drive exhaustion genes, or (3) NFAT is “partner-less” at the promoter, due to its overactivation (Figure 3) [41,42]. Intriguingly, it has recently been shown that forced expression of C-JUN can reverse the exhaustion phenotype in epigenetically exhausted/dysfunctional cells, highlighting the new therapeutic opportunities of manipulation of key transcriptional regulators [42]. 

It remains a subject of debate the degree to which intra-tumour exhausted/dysfunctional T-cells are induced and maintained by chronic TCR or CAR-mediated sustained antigen stimulation versus suppressive effects of the tumour microenvironment. Paradoxically, since the effector cell pathway of immune responses to infection is physiologically short-lived and dependent for an ongoing response on recruitment of new effectors from a population of memory cells, effector cells might not be the optimal population for the successful eradication of disease in the solid tumour setting. Engineering approaches to invigorate T-cells in the exhaustion pathway may prove more effective. 

### 1.2. Defining CAR-T-Cell Exhaustion and Dysfunction

Whilst there is growing consensus that T-cell exhaustion and dysfunction are key concepts in the CAR-T field and especially in the solid tumour context, there is much ongoing debate regarding the definition of these terms in normal T-cell physiology and in CAR-T biology. In this review we will use the term “dysfunction” deliberately loosely to describe two limitations on CAR-T behaviour: (1) The developmental pathway of exhaustion referred to above and (2) the sense of having misplaced effector function; i.e., on-target off-tumour activity against healthy tissues sharing common antigens. It is to understand and overcome these limitations using engineering approaches that is our subject in hand. 

## 2. Engineering Solutions

### 2.1. Chimeric Antigen Receptor and Vector Design

An optimal CAR will specifically bind an antigen expressed exclusively in tumour cells to form an effective immunological synapse leading to downstream T-cell signalling and a potent and specific anti-tumour effect. In first generation CARs, signal 1 of T cell activation is affected by a single signalling domain most commonly CD3zeta (CD3ζ), whilst second and third generation CARs have an additional one or two co-stimulatory domains, respectively (Figure 1B). Poor CAR design can lead to suboptimal T-cell activation, on-target off-tumour toxicities and antigen-independent (tonic) signalling; the latter resulting in an exhausted phenotype [43,44,45,46,47,48]. All regions of the CAR are important in determining function, and hence, are substrates for engineering solutions (Figure 1C). 

#### 2.1.1. Antigen Binding Domain

The canonical ectodomain is a single-chain antibody variable fragment (scFv) derived from a monoclonal antibody. CARs with lower affinity ScFv demonstrate higher discrimination between cancer cells and normal cells when the target antigen is expressed at lower density in normal tissues. Chmielewski et al. have reported that T-cells with low-affinity scFv (K_d_ > 10^−8^) were activated only upon recognition of target T-cells with high ErbB2 density whilst high affinity (K_d_ < 10^−8^) did not discriminate between low and high density targets [49,50,51]. Park et al. suggest that CAR-T-cells with a micromolar affinity (~10 μM) against ICAM-1 demonstrated superior anti-tumour activity and higher dependency on antigen density compared with nanomolar affinity (1–100 nM) [52]. Restricting binders to high affinity, therefore, risks enhancing on-target off-tumour toxicity, whilst there is emerging data that lower affinity receptors, for example, with slow off-rate kinetics, can be less toxic with equivalent efficacy [53,54].

In addition to the risk of on-target off-tumour toxicity, the scFv domain has been associated with CAR tonic signalling. Long and colleagues have shown that second generation CAR-T-cells with scFvs against GD2, CD22 and ErbB2 and with a CD28-CD3z endodomain, in contrast to CD19 CARs, signal constitutively during their in vitro expansion [43]. These constitutive active CARs had enhanced T-cell exhaustion and diminished anti-tumour activity in vivo. The authors demonstrated that scFv clustering, due to interaction between scFv framework regions, was responsible. Similarly, Frigault et al. have shown that high level expression of certain ScFv-containing CARs leads to long term antigen-independent signalling and functional exhaustion [44]. 

#### 2.1.2. Spacer and Transmembrane

The spacer, together with a transmembrane domain forms a bridge between the CAR’s signalling domain and the binding domain. The length and flexibility of the spacer can affect the ability of the CAR to form a functional immunological synapse, and the optimal design is likely to be affected by the size of the targeted antigen and position of the epitope. Commonly used spacers include CD8a, CD28 proximal ectodomain, and the IgG-derived Fc CH2/CH3 and hinge domains (Figure 1D). IgG–derived spacers, either IgG1 or IgG4, can result in off-target T-cell activation, at least in part as a result of their binding to Fc gamma receptors (FcγR) present in innate immune cells [46,47,53,55,56]. Hombach et al. showed that binding to the CH2 CH3 domain of the IgG1 spacer by FcγR led to the secretion of cytokines by NK cells and monocytes. This interaction was reduced upon introduction of two distinct mutations in the PELLGG and ISR motif in the CH2/CH3 domain [47], and in turn, restored CAR-T cytotoxicity and persistence in vivo [48,53]. Similarly, Almåsbak and colleagues demonstrated that this IgG1 spacer-FcγR interaction could induce severe toxicities [56]. Watanabe and colleagues further demonstrated that the IgG1 CH2/CH3 domain in a PSA-specific CAR was responsible for tonic CAR T-signalling and subsequent functional exhaustion [46].

A transmembrane sequence links endo and ectodomains. Most CAR structures have incorporated the transmembrane of one of the adjacent ectodomain or endodomain sequences (e.g., CD8 or CD28: Figure 1). A recent study by Majzner et al. has demonstrated the potential importance of transmembrane sequences, particularly for targeting low-antigen-expressing targets [57]. Directly comparing the CD28 Hinge/transmembrane with CD8 hinge/transmembrane, CD28 reduced the activation threshold through the generation of a more stable immunologic synapse. The relative contributions of transmembrane and stalk are still to be determined, but these observations might provide insight into the relative function of two FDA approved CD19 CAR-T products; siagenlecleucel (CD8 S/TM) and axicabtagene ciloleucel (CD28 S/TM) (Figure 1B).

#### 2.1.3. Signalling Domain

During the process of CAR immune synapse formation following ligand binding the three immune-receptor activation motifs (ITAMS) present in CD3zeta domain become phosphorylated (signal 1) by Lck leading to the recruitment and activation of downstream Zap70 and LAT. Multiple studies have shown that a co-stimulatory signalling domain/s incorporated into second generation CAR is required for optimal CAR-T-cell activation. The most widely used co-stimulatory signalling moieties are of the B7 superfamily (e.g., CD28, ICOS) and TNF superfamily (4-1BB and OX40), which have distinct effects on CAR-T-cell phenotype, metabolism, and pharmacodynamics [58]. CD28-CD3z CAR-T-cells have shorter persistence compared with 41BB-CD3z. CD19-CD28-based CAR-T-cells typically persist between three to eight weeks post-infusion, while 4-1BB CAR T-ceIls have been detected for years [59,60]. Interestingly the two FDA approved CD19 CAR-T-cell products with the FMC63 binder, siagenlecleucel (41BB-ζ) and axicabtagene ciloleucel (CD28-ζ) are both capable of inducing sustained remissions in leukaemia despite different kinetics. 

T-cell activation kinetics downstream of CAR signalling is likely to be dependent on the integration of factors that affect signal strength involving the ectodomain/transmembrane initiation of signalling, and the endodomain signalling motifs themselves which affect signalling both qualitatively and quantitatively. CD28-CD3z CAR-T-cells elicit a faster and more potent response associated with predominant glycolytic metabolism and are characterised by a more differentiated phenotype. In contrast, 4-1BB-CD3z CAR-T-cells have a more memory-like phenotype and increased persistence associated with a metabolic shift to beta-oxidation and mitochondrial biogenesis [61]. In a phosphoproteomics study, Salter et al. confirmed that CD28-based CAR-T-cells react with faster kinetics and intensity than 4-1BB-based upon activation, but triggered identical protein phosphorylation events. Additionally, 4-1BB CAR-T-cells were characterised by a memory-like phenotype, while CD28 CARs presented increased production of pro-inflammatory cytokines, increased exhausted phenotype, poor persistence and a more effector-like transcriptional profile. These observed characteristics of CD28 CARs were partially or completely abrogated by mutations at the Lck binding site of CD28 confirming the mechanistic link between CD28 signal and CAR-T-cell cell phenotype [62]. The mechanistic link was further elaborated by the demonstration that a single asparagine to phenylalanine mutation in YMNM in CD28, inhibits ERK signalling with an associated diversion to AKT signal and decreased calcium influx, abrogates the T-cell exhaustion phenotype, and improves anti-tumour control. The mutation in the YMXM motif presents an engineering solution for reducing exhaustion of CD28-co-stimulted CAR-T-cells. Interestingly, CAR-T-cells containing an ICOS co-stimulatory domain have increased persistence in comparison with CD28 CARs. Both ICOS and CD28 share the same YMXM motif, with the difference that the third amino-acid is phenylananine and asparagine, respectively. Thus, superior persistence of ICOS-based CARs may be attributable to the presence of this phenylananine within YMXM.

Others have generated second generation CAR constructs with ICOS as the chosen co-stimulatory domain [63]. In preclinical studies, the presence of ICOS drives CAR-T-cells into a T_H_17/T_H_1 differentiation state characterised by increased IL-17 and IL-22 production, enhanced persistence and improved anti-tumour function compared with CD28 or 41BB [64,65]. In one study of note, CD4 T-cells expressing CD28-based CARs have a T_H_1/T_H_2 phenotype a, while 41BB-CARs were more polarised to T_H_1 cytokines and away from T_H_17 and T_H_2. Moreover, combining ICOS and 4-1BB in third generation format further enhances in vivo activity of CAR-T-cells [65]. In addition to 4-1BB, CD28, and ICOS others have explored the potential use of OX-40 and CD27 for delivering co-stimulatory signals [58,63,66,67,68]. 

#### 2.1.4. Regulating CAR Expression and Function

Studies in TCR signalling indicate that avidity thresholds govern the successful generation of a functioning immune synapse. Hence, for lower affinity receptors or low target antigen density, higher expression of CAR will be required for successful T-cell triggering. However, simple overexpression of the chimeric receptor increases the risk of CAR-T dysfunction, due to the leakiness of many CARs accompanied by exhaustion.

To investigate the contribution of CAR expression levels in a tonic signalling Frigault et al. have used viral vectors of promoters with different potency to drive the expression of a c-Met CAR [44]. EF1a promoter resulted in bright CAR surface expression accompanied by non-antigen specific signalling, while this was not observed using CMV or truncated PGK promoters. Importantly, this observation with EF1a promoter was made using FMC63-based CD19 CAR previously described as free from tonic signalling [43,45]. Eyquem and colleagues further showed attenuation of CAR expression by targeting the CAR into the T-cell receptor alpha constant (TRAC) locus [69]. Driving CAR surface expression by the TRAC promoter resulted not only in evenly distributed expression levels, but also in reduced tonic signalling and superior anti-tumour activity [69]. 

The potency of CAR-T-cell activation can be further regulated to diminish downstream exhaustion by titrating the number of ITAMs in CD3ζ [57,70]. Wild type CD19-28ζ CAR (containing three ITAMs) was compared with a panel of CARs with deletions resulting in one or two ITAMs. Results from in vivo stress experiments revealed that single-ITAM CARs outperformed the unmodified versions, but interestingly this benefit was restricted to the retention of the single ITAM in the membrane-proximal position [70]. Although the reduced signal strength appears to be beneficial, recent work from Majzner and colleagues have demonstrated that it results in poor efficacy against low-antigen expressing tumours. Thus, increasing signalling may be a requirement targeting malignancies with lower antigen density [57,71].

Taken together, the tonic signalling and associated differentiation and functional exhaustion phenotype are driven by a number of factors, including ScFv, transmembrane and stalk, signalling endodomain, expression levels and kinetics, and target antigen density. One potential solution for integrating this number of variables to identify an optimal structure is through a library screening approach. For example, an scFv phage library was used to identify an optimal BCMA-specific binder which resulted in the generation of a highly effective CAR [72]. In a proof-of-concept study, Alonso-Camino et al. have used CARbodies; scfVs fused to a first generation CAR structure. Upon multiple rounds of selection-activation against CEA-expressing HeLa cells, they were able to identify dominant scFV-CAR clones [73].

### 2.2. Optimising Manufacturing Steps

#### 2.2.1. Cell Product Selection Steps

The question of which T-cell subset/s should be selected for CAR-T therapy to maximise persistence and function has been evaluated for many years. Both CD8+ cytotoxic T-cells and CD4+ T helper cells play important roles in anti-tumour immunity, with the assumption that CD8+ T-cells are primary effectors because of their enhanced cytotoxic properties, and indeed, as a purified cell product are capable of tumour elimination [74]. Liadi et al. amongst other have demonstrated that CD4+ CAR-T-cells can participate in killing, but do so at a lower rate than CD8+ CAR-T-cells—most likely due to the lower Granzyme B content [75]. CD4 helper cells moreover play an essential role in the promotion of cytotoxicity, and hence, it has been argued, and implemented in clinical studies, that an equal ratio of CD4 and CD8 is optimal for adoptive transfer [76]. 

The field has broadly identified the need for T-cell persistence to effect long term cancer response and the recognition that this is dependent on memory T cells [77]. The classic description of differentiation describes a transition from naïve (T_N_) to central memory (T_CM_) to Effector memory (T_EM_) to terminally differentiated or TEMRA (T_EFF_). Of memory cells, T_CM_ expressing CD62L+ and CCR7+ home to lymph nodes can promote memory responses whilst having relatively low immediate effector function [78]. In contrast T_EM_ and T_EFF_ cells lose expression of CD62L and CCR7, express KLRG-1 as a marker of terminal differentiation and preferentially migrate to tissues and sites of inflammation providing essential effector function of CAR-T-cells against tumours [79,80]. Whilst it is generally accepted that an optimal CAR-T-cell product is one that is armed with immunological memory and capacity to generate new effectors at disease sites, the extent to which these memory cells might correspond with precursor cells in the effector or exhaustion developmental pathways (T_EFF_ and T_EX_ precursors: Figure 2) is yet to be determined. However, the capacity to maintain effector function in the face of ongoing antigenic stimulation appears a particular property of the exhaustion pathway. Indeed, Miller and coworkers have recently shown that progenitor exhausted TILs are more effective than terminally exhausted TILs at effecting tumour control in mice [27]. 

The classic model of differentiation has also been refined by the recognition of memory stem cells (T_SCM_) sitting in the differentiation continuum between naïve and central memory subsets [81]. However, understanding of stemness and memory has now been complemented by the insights of parallel exhaustion and effector pathways deriving from naïve cells following antigenic stimulation (Figure 2). The recognition that the exhaustion pathway might be more optimal for sustained T-cell response in the face of chronic antigen stimulation, means that much of the earlier work on pre-selection of optimal memory cell subset for adoptive transfer, with a strong data set favouring central memory or naïve selection [79,80,82,83,84,85], can now be re-evaluated to incorporate exhaustion versus developmental pathways [19]. 

Selection of rare populations, such as T_SCM,_ precursor T_EX_, etc., poses particular problems when seeking to generate a clinical T-cell product at scale. One solution recently proposed by Klebanoff et al. is to exploit the capability of memory cells to influence T_N_ differentiation during priming which occurs through nonapopototic FAS signalling, resulting in the activation of Akt and S6 responsible for cellular differentiation and metabolism. They found isolation of T_N_ cells before priming by memory T-cells, or blockade of Fas signalling, prevented memory T-cell induced differentiation and preserved the anti-tumour efficacy of T_N_ cells [86] 

#### 2.2.2. Activation Methods

Most CAR-T-cell production protocols incorporate initial T-cell activation to promote expansion and facilitate viral transduction. Optimal activation should lead to sufficient T-cell expansion without causing T-cell differentiation and activation-induced cell death (AICD) [87]. Most commonly, CAR-T-cell products are activated using anti-CD3 and anti-CD28 monoclonal antibodies, or antibody-coated magnetic beads. Some evidence indicates CD3/CD28 beads induce higher cytokine production than anti-CD3 monoclonal antibody and IL-2 [88]. Furthermore, beads activation is reported to induce the generation of less differentiated and potentially less senescent CAR-T-cells with enhanced proliferative capacity and early in vivo anti-tumour responses [89]; they are commonly used in clinical protocols associated with high CAR-T response rates [6].

The use of retronectin during activation can increase retroviral transduction efficiency and can promote T_N_ and T_SCM_ phenotype and increase the amount of cytotoxic CD8+ T-cell component of the cell product [90,91,92]. Retronectin, however, comes with some health warnings: Specifically reported reduced cytokine secretion and stimulation of malignant B cells in the patient T-cell product [92]. Several alternative activation methods are competing for clinical implementation; for example, a specialised polymeric nanomatrix conjugated to humanised recombinant CD3 and CD28 agonists in a serum-free media which is reported to increase T_CM_ cells with higher IL-2 secretion and lowered exhaustion marker expression [91]. Since T-cell stimulation to facilitate transduction leads to a degree of differentiation and loss of memory/precursor populations (irrespective of exhaustion or effector lineage) there is interest in manufacturing approaches that do not involve in vitro expansion, such as non-viral transfection of large blood volumes with expansion in vivo [93,94] or generation of CAR-T from induced pluoripotent stem cells, although the latter has not translated to the clinic [95]. 

#### 2.2.3. Ex Vivo Expansion Milieu

Interleukin(IL)-2, IL-15, IL-7 and IL-21 are members of a cytokine family whose heteromeric receptors share the common gamma (γ) chain (γ_c_) [96,97]. Expanding T-cells have a homeostatic requirement for γ_c_ cytokines, and reduction of competition for these cytokines is thought to be key in the increased survival of transferred T cells following lymphodepletion preconditioning [98]. Hence, the addition of γ_c_ cytokines during the CAR-T-cell production process or has been widely studied for composition, quality, and phenotype of the final product [87]. Each γ_c_ cytokine has been evaluated for augmentation of T-cell anti-tumour immune responses, building on the early work with tumour infiltrating lymphocyte expansion, performed in the presence of high concentration IL-2 [99]. IL-2 has been widely added to culture conditions for the clinical product CAR-T-cell expansions, however repetitive stimulation with IL-2 can result in T-cell exhaustion, reduced persistence and IL-2 dependency [100], which can be addressed by engineering IL-2 autonomy through local secretion by the CAR-T [101]. Moreover, IL-2 plays a crucial role in the development and maintenance of inhibitory regulatory T-cells (T_regs_) [102,103]. Expansions of CAR-T-cells utilising high levels of IL-2 followed by cell expansion drives cells to become terminally differentiated and exhausted with compromised ability to successfully enter the long-lived memory pool [104,105]. Several groups have investigated combinations of IL-7, IL-15 and IL-21 γ_c_ cytokines during expansions and all have been shown to preserve a less differentiated state although several different combination approaches have been evaluated and there is no single unified protocol [87,90,106,107,108,109].

The retention of the T_SCM_ population is particularly interesting, since its presence in a T-cell product correlates with sustained T-cell persistence and persistence of tumour control [81,104,110]. Of note, Cieri et al. and Gattinoni et al. using similar approaches to culture T-cells from naïve precursors showed the addition of IL-7 and IL-15 following bead stimulation enhanced development of the T_SCM_ phenotype [81,107], whilst Schmueck-Henneresse et al. found higher numbers of T_SCM_ and T_CM_ with IL-17 and IL-15 compared with IL-2 expansion [111]. As a single agent cytokine for T-cell culture, IL-15 has also been shown to reduce exhaustion marker expression, increase anti-apoptotic properties, and improve proliferation and preservation of the T_SCM_ phenotype compared to IL-2 expansion [112], with reduction of mTORC1 activity postulated to be a major mechanism [100]. IL-21 has also been reported to suppress antigen-induced differentiation of CD8+ T-cells, and enhance anti-tumour activity when compared to IL-2 and IL-15 [113].

#### 2.2.4. Drugs to Regulate T-Cell Development

The supplementation of pathway inhibitors during ex vivo T-cell expansion might cause an interruption of T-cell differentiation by the inhibition of specific signalling pathways, and thus, shift the T-cell phenotype in the final CAR-T-cell product towards a less differentiated state [114]. Gattinoni et al., demonstrated that the induction of the wnt-beta-cantenin (WNT-β catenin) signalling pathway during expansion by inhibitors of glycogen synthase kinase-3beta (GSK3β) blocked T-cell differentiation and promoted the generation of self-renewing T_SCM_ cells with an enhanced proliferative and anti-tumour capacity [115]. Similarly, it has been shown that inhibition of phosphatidylinositol 3-kinase δ (PI3Kδ) and antagonism of vasoactive intestinal peptide (VIP) signalling during expansion partially inhibits T-cell terminal differentiation [116]. Inhibitors of the mTOR pathway, such as rapamycin, have been used in preclinical settings to increase the number of memory T-cells by inhibition of further differentiation [117]. Interestingly, recent data identifies that IL-15 benefits on T-cell function are imparted through its effects on diverting T-cell from mTOR to STAT5 signalling with metabolic switch away from glycolysis and towards increased mitochondrial oxygen consumption rate [100].

### 2.3. Transcriptional Reprogramming to Mitigate or Exploit Exhaustion

New insights have identified at least two parallel T-cell developmental pathways (exhaustion and effector) downstream from T-cell reactivation, and have listed a number of transcription factors capable of functioning as master regulators. This new understanding presents both a challenge and an opportunity for rewiring CAR-T-cells for sustained effector function in the context of repeating ongoing antigenic stimulation. TCF-1 is expressed in precursor cells of both exhaustion and effector pathways, raising the hypothesis that promoting its expression might sustain CAR-T function in vivo [25,34,37]. Shan and co-worker have recently demonstrated that ectopic Tcf-1 expression expands stem cell-like T_EX_ cells leading to improved anti-tumour control [118]. The TOX transcription factor both establishes and maintains the exhaustion pathway, suggesting that its inhibition in CAR-T-cells might promote the effector pathway with its attendant higher cytokine and killing, albeit potentially short-lived responses [19]. Working with chimeric antigen receptors in knock out mice for the *tox1* and *tox2* genes, Seo et al. have shown that CAR-T deficient for *tox1* and *tox2* are more effective than wild type in suppressing tumour growth [33], whilst in a TCR transgenic adoptive transfer model, Khan and co-workers showed that heterozygous deletion of TOX1 in T-cells increases tumour growth control [31]. However, since effector pathway T-cells are reported to be prone to activation-induced cell death, it will be important to evaluate TOX deficiency in CAR-T in the context of hard to treat solid tumour models. Similarly, Chen and co-workers have evaluated CAR-T-cell function in T-cells derived from mice deficient in the genes encoding NR4A-1, 2 and 3; NR4A similar to TOX is induced downstream from NFAT signalling, and its sustained expression is promoted by TOX (Figure 3) [33,40]. Against a solid model, NR4A deficient CAR-T-cells have superior in vivo efficacy, suggesting that exhaustion pathway involvement is not essential for CAR-T-cell function, at least in short term efficacy studies [39]. In addition to its role initiating the exhaustion pathway, TOX maintains the exhaustion phenotype by epigenetic mechanisms [32,119]. Intriguingly however, Lynn and co-workers have recently shown that functionality can be restored to epigenetically imprinted exhausted cells by forced expression of c-jun in CAR-T-cells; the mechanism being to promote transcription of NFAT-AP1 complexes and shift the transcriptional balance towards effector cells (Figure 3) [42].

### 2.4. Using Boolean Logic and Switch Systems to Limit Stimulation and Mitigate Exhaustion

The concepts of Boolean Logic gates have been applied to CAR-T-cell design primarily as a tool for increasing specificity through integrating signals from two or more target antigens (Figure 4). Thus, far studies have focused on AND gates (two target antigens requires for full activation as a means to limit toxicity), OR gates (two equally powered CARs against alternate target antigens in the same tumour) and NOT gates (an off signal associated with a target antigen present on normal tissue as a means to limit off-target toxicity). 

Several paired target antigens for evaluation in specific tumour types have been evaluated using AND gate approaches in preclinical models. Specific pairs of antigens include ErbB and MUC1 in breast cancer [120], mesothelin and folate receptor in synthetic target cell lines engineered to express one or both antigens, and PSMA/PSCA in a prostate cancer model [121]. These AND gate approaches all demonstrate proof of concept that avoidance of off-target toxicity with the maintenance of anti-tumour activity can be affected, but fine-tuning is required in terms of relative affinities of the binders and optimal spacer lengths. The CAR with an endodomain providing signal 2 but no signal 1 is also referred to as chimeric co-stimulatory receptor (CCR) and is said to provide “co-stimulation in trans”. A CCR providing co-stimulation in trans enhances proliferation and survival, but has no effect on cytotoxicity [121,122]. 

To elaborate on the OR gate concept, Grada and colleagues introduced a novel design of a single chimeric antigen receptor which encompass specificity for two distinct antigens [123]. This bispecific tandem receptor, named as TanCAR, has two distinct scFvs connected through a glycine-serine linker. Recognition of either of the two antigens, for example, HER2 or IL13Rα2, results in CAR-T-cell activation, while simultaneous antigen binding has a synergistic effect [123,124]. Several groups have further designed bi-specific CARs, which are utilising the OR-gate circuit, with examples being the “OR-gate CARs” targeting CD19/CD20 and “nanoCARs” specific for HER2 and CD20 [125,126]. 

By applying the NOT gate logic in the cell level, CARs coupled with an immune-inhibitory signal (iCAR), such as PD-1 or CTLA-4, have been generated [127]. The iCAR, which recognises an antigen expressed on normal cells, is co-expressed with an activating CAR A with specificity towards a tumour-associated antigen. CAR-T-cell response, thus, is only allowed when CAR A engages with its associated antigen in the absence of antigen B. This immune-blocking effect is temporary as signalling CAR-T-cells are able to elicit immunological response upon subsequent encounter with the antigen expressed on malignant cells [127].

#### Beyond Basic Logic; Sequential Logic Gates and Molecular Switches

The basic logic principles have been extended to the concept of the sequential AND gate, whereby engagement with one target antigen triggers the expression of a CAR against a second target antigen (Figure 4). Since the first antigen does not provide activatory T-cell signals, and the CAR against the second antigen is not expressed until the T-cell reaches the tumour. This arrangement can effectively limit both tonic signalling and exhaustion, as well as off-target toxicity. The most well-described approach involves synthetic notch receptors which on engagement with the first target antigen release a transcriptional activator protein sequestered into the receptor to allow expression of the CAR against the second antigen [128,129,130] (Figure 4D). 

Srivastava and colleagues further explored the clinical relevance of the concept of AND-gate synNotch circuits by using a model of CAR on-target off-tumour toxicity [131]. In this model, treatment of preconditioned mice with ROR1-targeting CAR-T-cells led to lethal splenic and bone marrow toxicity which was associated with the expression of ROR1 in stromal normal cells. In an effort to eliminate these effects, the anti-ROR1 CAR was placed under the control of a synNotch receptor specific for either EpCAM or B7-H3 as both of these molecules are reported to be absent from stromal cells. Expression of anti-ROR1 CAR was induced only upon engagement of EpCAM or B7-H3 to the syNotch receptor, thus only dual positive tumour cells were eliminated, and on-target off-tumour toxicities were prevented. Nevertheless, protection from toxicity was provided only when the ROR1-positive tumour cells where spatially separated from ROR1-positive normal cells as toxicities where observed when tumour cells and normal cells were co-localized [131].

Several other mechanisms exist for transcriptional induction in the tumour environment, including the use of soluble ligands that can be administered to the patient to switch on a silenced CAR in situ. For example, use of tetracycline to induce transcription at minimal promoters upstream of the CAR sequence through binding to tet regulators fused to transcriptional activators to trigger transcription from promoters. Alternate technologies can regulate the degradation of CARs. For example, the fusion of CAR to epitopes that target for proteosomal degradation can be reversed by the addition of competitive inhibitors [132]. All these approaches, through limiting CAR expression or directing expression to the site of disease, have the potential to mitigate both exhaustion and off-target toxicity.

A particular refinement of this approach from our own laboratory makes use of gamma delta T-cells that naturally can distinguish cancer cells from corresponding normal tissue because both the TCR and other innate receptors sense cell surface molecules associated with stress and danger to activate the T-cell. By expressing chimeric co-stimulatory receptors against target antigens to provide co-stimulation in trans, the engineered cells can mount effective responses against tumour cells expressing the target antigen whilst ignoring normal tissues expressing the same target antigen at the same levels [133]. Indeed, single cell signalling studies performed with this model system has demonstrated a lack of tonic signalling, which is linked with the absence of expression of exhaustion markers [45]. 

The concept of preventing exhaustion through limiting CAR expression need not be dependent on complex engineering steps. Mackall and co-workers have demonstrated that the RTK inhibitor dasatninib inhibits CAR-T signalling sufficiently during manufacturing to reduce exhaustion from tonic signalling and enhance in vivo efficacy. This approach is particularly attractive, since dasatinib is an approved drug with anti-cancer properties that can both be added to manufacture and administered to the patient post-adoptive transfer. Similar approaches could be considered with agents such as PI3K and ERK inhibitors, which have been shown to have similar beneficial effects on CAR-T-cell function in vitro. 

### 2.5. Creating A Pro-Inflammatory Host/Tumour Microenvironment

Equipping CAR-T-cells to retain the capacity to proliferate overcomes an intrinsic limitation of their biology and specifically the requirement to maintain proliferation in the presence of repeated antigenic stimulation. However, the solid tumour microenvironment presents an array of barriers specifically to inhibit sustained T-cell responsiveness. Efforts to target this limitation have focused on blocking tumour inhibition and/or creating an alternate pro-T-cell inflammatory milieu at the tumour site. 

#### 2.5.1. Administering Cytokines

Building on the insights from the culture in the presence of γ_c_ cytokines, investigators have evaluated the addition of cytokines as drugs, given to the host following T-cell adoptive transfer [112]. Early studies in tumour infiltrating lymphocytes expanded in IL-2 involved the co-administration of high dose IL-2, which was deemed to be essential for their persistence following adoptive transfer [134]. However, IL-2 has many drawbacks; namely, significant toxicity at high dose, and induction of regulatory T-cells at a lower dose [102,135,136], which has been mitigated through depletion of CD4 cells, likely to limit T-cell persistence [135]. Moreover, IL-2 administered in the context of lymphopenia has enhanced potential for Treg induction, limiting its scope in the CAR-T field [137]. Similar concerns about toxicity relate to other cytokines, with IL-12 being highly toxic [138], although recent studies with continuous infusion IL-15 show significant promise [85]. Engineering of CAR-T-cells to produce larger amounts of cytokine locally is a potential solution to the question of toxicity and delivery.

#### 2.5.2. Engineering Cytokine Signalling

Leen et al. initially described the concept of inverted cytokine receptors (ICR) in which the ectodomain of IL-4 was fused to the signalling domain of the immunostimulatory IL-7 receptor (4/7 ICR) [139]. IL-4 has abundant expression within the tumour microenvironment and is known for its immunosuppressive properties, while IL-7 stimulates the production of T_H_1 pro-inflammatory cytokines, thus by this approach an inhibitory signal is converted to immunostimulatory. Several groups have co-expressed an ICR with a CAR [140,141]. For example, IL-4/IL-7 ICR [140] or IL-4/IL-21 [141] co-expressed with the second generation to promote T_H_1 and T_H_17 properties, respectively, on engaging target antigen, and in the case of IL-21, reducing exhaustion markers expression when compared to the experimental controls. An alternate approach is to develop CAR-T-cells that constitutively secret and deliver cytokines, such as IL-12, IL-18 and IL-15, or other anti-cancer proteins [142,143,144,145,146,147].

IL-15 exists in both membrane-bound and secreted form and the former signals by both autocrine and paracrine pathways when co-expressed with a CAR. Hurton and colleagues thereby generated CAR-T-cells co-expressing a tethered IL-15 chimeric receptor (mbIL15) characterised by increased T_SCM_ phenotype, expression of TCF-1 and longer in vivo persistence [146]. CAR-T-cells expressing the mbIL15 appeared to have phenotypical characteristics and express molecular markers similar to T_SCM_. Interestingly, CAR-T-cells persisting more than 65 days upon removal of the antigen stimulus had a transcriptional profile associated with early stages of differentiation and were characterised by elevated TCF-1^63^. An elegant alternate approach to engineered IL-15 signal is to modify the endodomain of a second generation CAR to include IL-2R beta chain (provides equivalent of low-affinity IL2R signal, which is functionally similar to IL-15 signal) and a modification of CD3ζ domain to incorporate the YXXQ motif, which provides the equivalent of STAT3 signal. Hence, in the presence of the target antigen, this receptor provides T-cell activation, co-stimulation and cytokine signal, which together prolong survival and efficacy of transferred T-cells [148].

#### 2.5.3. Targeting Immunosuppressive Mechanisms

Innumerable mechanisms of tumour immune evasion by suppressive mechanisms exist and have recently been reviewed elsewhere [149]. Amongst the targetable evasion mechanisms of particular focus are TGF-beta and PD1 pathways. For example, Kloss et al. reported that blockade of TGF-beta through co-expression of an anti-PSMA CAR with dominant-negative TGFBRII resulted in improved anti-tumour efficacy and enhanced CAR-T-cell proliferation [150]. For PD1 targeting, in addition to the NOT gate approaches with decoy receptor listed previously [127], the approaches have been by either gene disruption by using the CRISPR/Cas9 technology or by combination therapy with a checkpoint inhibitor and a CAR-T-cell therapeutic [151,152,153,154,155,156]. Although the reported clinical experience of combining CAR-T-cells with checkpoint inhibitor antibody remains limited, in a case study of a patient with refractory diffuse large B-cell lymphoma addition of pembrolizumab, a PD-1 blocking antibody, reversed refractory disease to affect clinical response [154]. Strategies to target PD1 at the genomic locus in combination with CAR expression in T-cells using CRISPR/Cas9 technology are beginning to be evaluated in the clinic [157] following preclinical proof of concept studies [152,153] and have the capacity for translation into universal CAR-T approaches [155]. 

## 3. Conclusions

One overarching concept is that both too much and too little signal can limit CAR-T-cell function (Figure 5). Multiple components of CAR-T genesis can influence overall signal strength and duration throughout the CAR-T life-cycle, and this can impact on effector function/dysfunction. Hence, in addressing the solutions, our discussion has been framed in the context of the starting materials, the CAR structure with its inherent, as well as antigen-dependent signal, and the effect of manufacturing conditions on the T-cell host. 

In this rapidly evolving field, fundamental insights into the regulation of strength and duration of T-cell immune responses will aid the intelligent design of the next generation of anti-cancer engineered T-cells and development of more off the shelf products, such as has been described in the universal CAR-T concept [158]. Many unanswered questions remain, for example, the relationship between T-cell signal strength, metabolic reprogramming and the ability to sustain prolonged effector function and proliferation. Nevertheless, learning from the new mechanistic insights of T-cell exhaustion versus effector pathways, the CAR-T-cell field is well-positioned to use engineering approaches to translate T-cell signalling into sustained T-cell effector function with enhanced proliferation and stem-like qualities. 

## Figures and Tables

**Figure 1 cancers-12-02326-f001:**
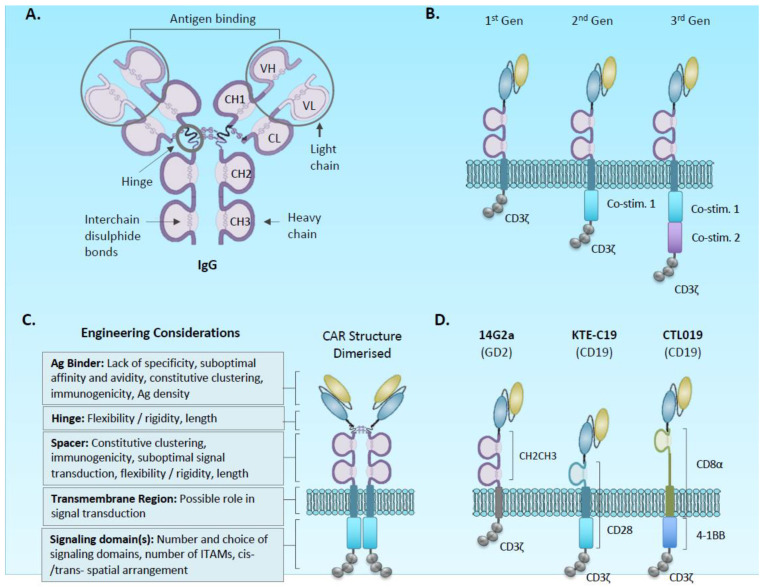
Structure of chimeric antigen receptors. (**A**) structure of antibody indicating hinge and CH2/CH3 domains incorporated into chimeric antigen receptor (CAR) spacer, and variable light and heavy chains incorporated into ScFv component of CAR. (**B**) prototypic first to third generation CAR structures incorporating the CH2-CH3-hinge spacer. (**C**) Example CARs used in clinical studies: The 14G2A anti-GD2 first generation was an early clinical study in solid cancer [11], whereas KTE-C19 and CTL019 are FDA approved agents targeting CD19. Note the different transmembrane regions derived from adjacent co-stimulatory domains. (**D**) CARs are thought to be expressed at the cell surface as dimers linked by disulphides within the spacers. Each region of a CAR presents an opportunity for engineering strategies.

**Figure 2 cancers-12-02326-f002:**
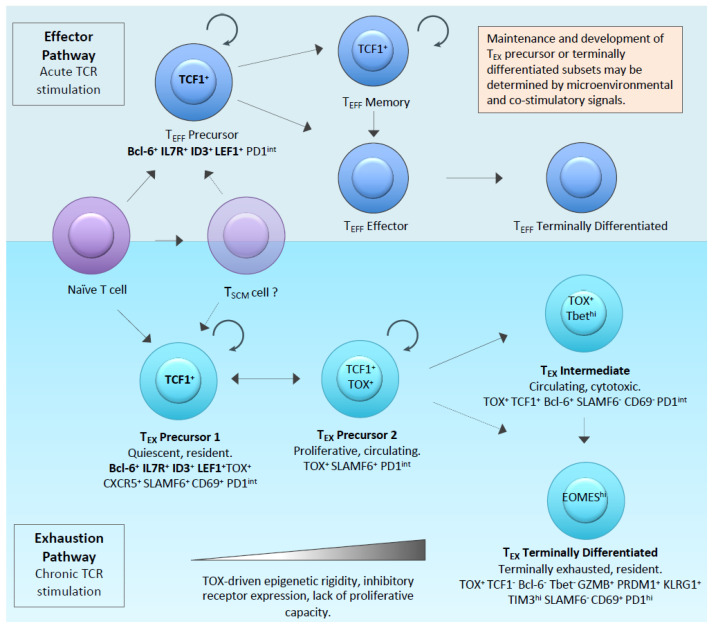
Overview of the emerging new understanding of parallel effector (**top**) and Exhaustion (**bottom**) developmental pathways. Key genes and surface markers that are currently used to define the respective putative differentiation states are shown. The markers that are common to precursor cells irrespective of effector or exhaustion pathways are shown in bold.

**Figure 3 cancers-12-02326-f003:**
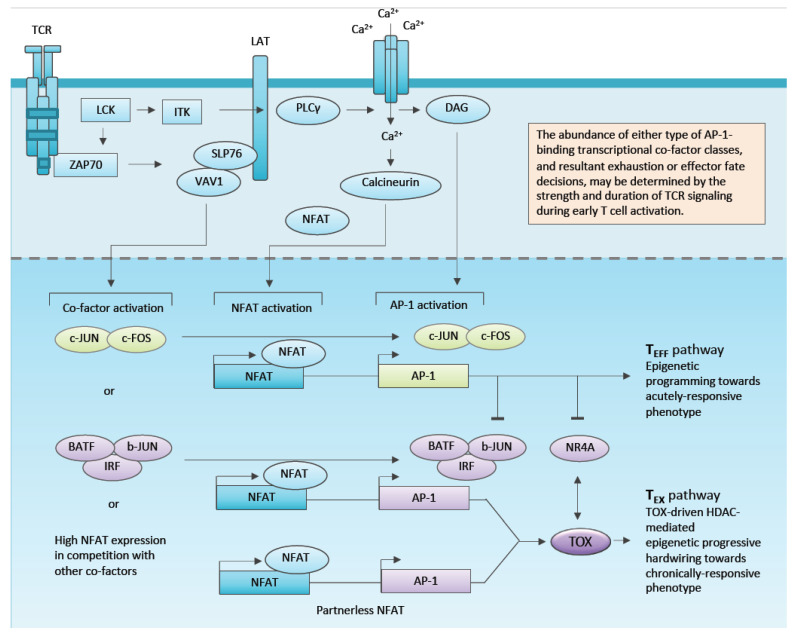
Emerging understanding of how T cell signal strength might determine T cell fate through integrating NFAT with AP1 transcription factors and regulating master transcription factor regulators, such as TOX. A canonical AP1 transcription factor is shown as c-JUN/c-FOS heterodimer. Oblong boxes represent consensus binding sites in promoters. The concept of “partner-less NFAT” is depicted as NFAT binding to its consensus with no AP1 family transcription factors bound to adjacent AP1 site. The extent to which this phenomenon is determined by high NFAT versus absence of AP1 binding transcription factors is not fully understood.

**Figure 4 cancers-12-02326-f004:**
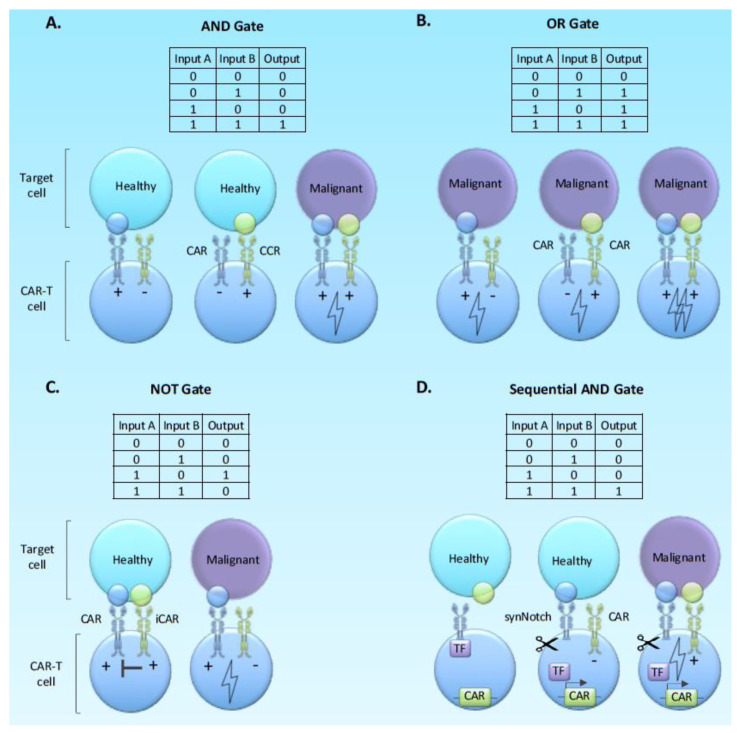
Boolean logic gate approaches to CAR-T cell engineering. Boolean logic is represented as both binary inputs and outputs and how this translates to a biological counterpart through on and off signals.

**Figure 5 cancers-12-02326-f005:**
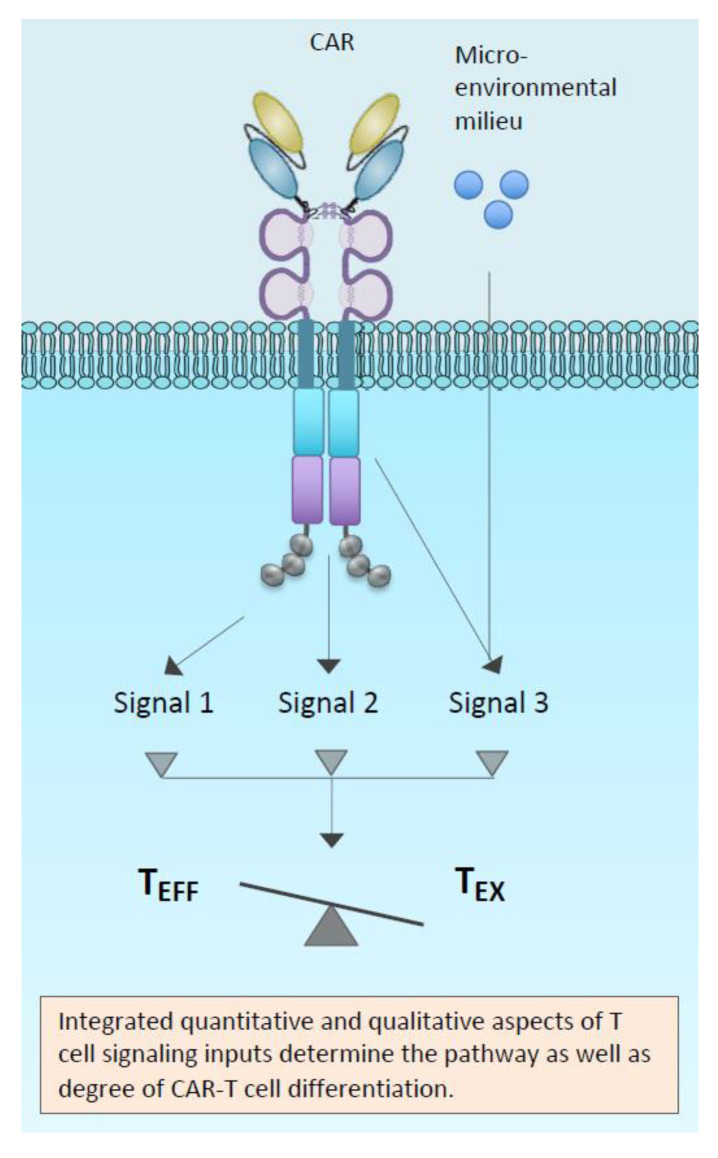
Emerging understanding of how both quality and quantity of signalling downstream from CARs might determine cell fate between exhaustion and effector pathways.

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
