# Peer review of "Engineering Solutions for Mitigation of Chimeric Antigen Receptor T-Cell Dysfunction"

_cancers, 2020, doi:10.3390/cancers12082326_

Round 1

Reviewer 1 Report

The manuscript brought to the attention of Cancer’s Editorial Office for publication is a well-conceived and written review that essentially covers an important area of interest in the field of CAR-T cell therapy for solid tumors. In their work Gavriil et al., highlight the importance of understanding the physiological role of T-cell exhaustion to ameliorate CAR-T engineering. In doing so, they brilliantly extricate in the complex field of cell signaling, from transcriptional to epigenetic regulation, determining fate and phenotypes of T Cells, with the intent of indicating the best T cell types candidate For Chimeric Antigen Receptor engineering.  

The entire work is carefully supported by the essential information derived from literature, ranging from system biology to developmental immunology and saliently put in frame and translated to discuss hurdles and limitations of CAR-T cell therapy for epithelial cancers.

The review describes two limitations on CAR-T behavior: exhaustion and misplaced effector function.

While dysfunction is exhaustively treated, also by virtue of the molecular portraits of the ICDs of Chimeric Receptors described so far, the importance of ECDs in regulating dynamics of the immunological synapse and immunosurveillance escape are, perhaps reasonably, less evaluated. A comment on “Universal CR” on the concluding remark section might have sense.  

I will suggest putting more effort in the figure legends, particularly Fig.2 and Fig.3, and, “possibly” Fig.4 for a general better comprehension.

Author Response

IN the revision we will significantly increase detail in the figure legends 2-4

Reviewer 2 Report

Authors present a high-quality and very well-written review that describes engineering solutions for mitigation of CAR-T-cell dysfunction. The manuscript is well structured, clear and easy to read. The figures are well-prepared and comprehensive.

The authors might consider citing the following reference when they write about CAR-T therapy-associated toxicities.
https://doi.org/10.1038/s41419-018-0918-x

Author Response

The suggested additional reference will be included in our revised manuscript

Reviewer 3 Report

Authors summarize the current model of T cell life cycle with respect to exhaustion and some strategies aiming at counteracting exhaustion in order to improve the T cell response in the long-term. This is a timely topic and of specific interest in the adoptive T cell therapy field, however, much data are summarized without focusing on the respective topic. The review will benefit from concentrating on concepts how exhaustion is induced and how it can be efficiently prevented. As it stands, unfortunately, major parts of the text resemble numerous reviews in the CAR field without providing novel impact.

Major comments

Fig 1A, lettering “scFv” is misleading since a scFv is a covalently linked VL-VH antibody and not a physiologic antibody region.

Fig 5 not very informative and can be deleted.

While Chapter 1 describes exhaution/dysfunction as a timely topic with respect to adoptive cell therapy of solid cancer. The first paragraphs of Chapter 2 summarize some engineering and production strategies generally applied when engineering a CAR, independently of addressing exhaustion. I strongly recommend to focus on the specific topic and to describe the impact on exhaustion/dysfunction. Chapter 2.3 is in line with the Chapter 1 topic. The link of Chapter 2.4 to exhaustion is less visible. AND and OR gating in the CAR field is well-known and does not need extensive presentation; however, the Chapter will benefit from demonstrating the link to exhaustion and strategies to improve T cell performance.

Minor comments

Line 67 should read “in chronic infection”

Author Response

We would like to thank reviewer 3 for detailed reading and thoughful suggestions.

Reviewer 3 however questioned the value of our article to the field by suggesting that there were many up to date review articles in the literature and therefore large sections of our article were redundant and could be deleted. We would like to rebut this suggestion by making the following argument:

Our review is covering the subject of CAR-T cell dysfunction in its broadest sense; both over functional and under-functional dysfunction. As far as I am aware there is no other review that covers this important balancing by bringing together the latest understanding of T cell dysfunction with the engineering approaches to control signaling and avoid both sides of dysfunction. Our aim is to present an educational article that brings the non-specialist up to speed with current understanding of T cell biology as it relates to CAR-T function, and is rooted in clinical application. We don’t see how the article can be shortened and still achieve those aims as per reviewer 3’s requests. The reviewer suggests that an article that covers just T cell exhaustion would be preferable but in our view that is missing the important key concept of balancing signalling through engineering to avoid both exhaustion and off target toxicity.

the following specific changes were suggested 

Fig 1A Scfv is misleading

This has been changed to “antigen binding”

Fig 5 is not very informative and can be deleted

Could we ask the editors to decide on this point. To our mind the

figure helps the non-specialist in particular to appreciate the

concept of balancing activation to mitigate exhaustion

Line 67 should read “in chronic infection”

We have changed it to read “response to infection” since both

acute and chronic infection are relevant here